# Role of Epicardial Adipose Tissue in Cardiovascular Diseases: A Review

**DOI:** 10.3390/biology11030355

**Published:** 2022-02-23

**Authors:** Michał Konwerski, Aleksandra Gąsecka, Grzegorz Opolski, Marcin Grabowski, Tomasz Mazurek

**Affiliations:** 1st Chair and Department of Cardiology, Medical University of Warsaw, 02-097 Warszawa, Poland; konwerski.mich@gmail.com (M.K.); aleksandra.gasecka@wum.edu.pl (A.G.); grzegorz.opolski@wum.edu.pl (G.O.); marcin.grabowski@wum.edu.pl (M.G.)

**Keywords:** atherosclerosis, cardiovascular diseases, epicardial adipose tissue, EAT, inflammation

## Abstract

**Simple Summary:**

Cardiovascular diseases (CVDs) are the leading causes of death worldwide. Epicardial adipose tissue (EAT) is one of the most important risk factors for cardiovascular events and a promising new therapeutic target in CVDs. Here, we summarize the currently available evidence regarding the role of EAT in the development of CVDs, including coronary artery disease, heart failure and atrial fibrillation; compile data regarding the association between EAT’s function and the course of COVID-19; and present new potential therapeutic possibilities, aiming at modifying EAT’s function. The development of novel therapies specifically targeting EAT could revolutionize the prognosis in CVDs.

**Abstract:**

Cardiovascular diseases (CVDs) are the leading causes of death worldwide. Epicardial adipose tissue (EAT) is defined as a fat depot localized between the myocardial surface and the visceral layer of the pericardium and is a type of visceral fat. EAT is one of the most important risk factors for atherosclerosis and cardiovascular events and a promising new therapeutic target in CVDs. In health conditions, EAT has a protective function, including protection against hypothermia or mechanical stress, providing myocardial energy supply from free fatty acid and release of adiponectin. In patients with obesity, metabolic syndrome, or diabetes mellitus, EAT becomes a deleterious tissue promoting the development of CVDs. Previously, we showed an adverse modulation of gene expression in pericoronary adipose tissue in patients with coronary artery disease (CAD). Here, we summarize the currently available evidence regarding the role of EAT in the development of CVDs, including CAD, heart failure, and atrial fibrillation. Due to the rapid development of the COVID-19 pandemic, we also discuss data regarding the association between EAT and the course of COVID-19. Finally, we present the potential therapeutic possibilities aiming at modifying EAT’s function. The development of novel therapies specifically targeting EAT could revolutionize the prognosis in CVDs.

## 1. Introduction

Cardiovascular diseases (CVDs) remain one of the leading causes of death worldwide [1,2], entailing enormous costs for healthcare systems [3]. Many of them could be avoided, as cardiovascular risk factors are largely reversible.

Epicardial adipose tissue (EAT) is defined as a fat depot localized between the myocardial surface and the visceral layer of the pericardium, and is a type of visceral fat [4]. Therefore, as is the case with abdominal obesity, EAT is one of the most important risk factors for atherosclerosis and cardiovascular events [5,6,7]. Moreover, the volume and thickness of EAT correlate with intra-abdominal fat mass and severity of obesity [8,9] and are independently associated with cardiovascular events [10]. Additionally, there are reports that EAT may be a new therapeutic target in CVDs [11,12]. 

The most common classification of the adipose tissue surrounding the heart includes (i) epicardial adipose tissue, (ii) pericardial adipose tissue, (iii) paracardial adipose tissue, and (iv) perivascular adipose tissue (Figure 1). Epicardial fat is located below the visceral pericardium. Pericardial fat consists of adipose tissues between the visceral and parietal pericardial layers and the fat depot on the external surface of the parietal pericardium. Paracardial fat involves fat deposits outside the parietal pericardium. The perivascular adipose tissue is a fat around the coronary arteries, irrespective of location [13]. 

There are many differences between EAT and other types of adipose tissue, including anatomical, histological, embryological, and genetic differences [14,15]. EAT is located between the pericardium and the myocardium [5] and is not separated from the myocardium and vessels by fascia, allowing paracrine or vasocrine effects [16] via cytokines and chemokines [17]. In health conditions, EAT has a protective function, including protection against hypothermia [18] or mechanical protection for the coronary circulation [19]. Additionally, EAT has an important role in energy supply to the myocardium [20]. Thanks to the ability to use free fatty acid (FFA) quickly, EAT may protect the myocardium from the cardiotoxic effect of a large amount of FFA [21]. The secretion of adiponectin from epicardial adipocytes is also an important function of EAT. Adiponectin protects coronary circulation, improves endothelial function, reduces oxidative stress, and indirectly decreases the level of interleukin-6 (IL-6) and C-reactive protein (CRP) [22,23,24]. However, under specific conditions such as obesity, metabolic syndrome, or diabetes mellitus, the protective properties may be destroyed and EAT becomes a deleterious tissue promoting the development of CVDs. The most data on the transition of EAT’s role is based on the observations of patients with coronary artery disease (CAD). For example, in one of our studies we showed a difference in gene expression in pericoronary adipose tissue in patients with and without CAD [25].

In this article, we summarize the currently available evidence regarding the role of EAT in the development of CVDs, including CAD, heart failure (HF) and atrial fibrillation (AF). Due to the rapid development of the COVID-19 pandemic, we also summarize data regarding the association between EAT’s function and the course of COVID-19. Finally, we present the potential therapeutic possibilities aiming at modifying EAT’s function in CVD. The role of EAT in the development of CVDs and COVID-19 is summarized in Figure 2.

## 2. Coronary Artery Disease

It has been shown that EAT and CAD are closely related at different levels: (i) in patients with CAD, the secretion of adipokines from EAT is altered; (ii) EAT has a proinflammatory features in patients with CVD risk factors and/or CAD; (iii) the amount of EAT and/or its proinflammatory state correlate with the severity of CAD and the instability of the atherosclerotic plaques; (iv) there is a relationship between EAT’s function and coronary microvascular dysfunction and artery spasm [14,17,26,27,28,29,30,31,32,33,34,35,36,37,38,39,40,41,42,43,44,45,46,47,48,49]. 

In patients with obesity, metabolic syndrome, or CAD, the epicardial adipocytes secrete less adiponectin and more leptin than in healthy people [26,27]. The decreased adiponectin expression attenuates endothelial function and leads to increased tumor necrosis factor-α (TNF-α) production, which increases inflammation and oxidative stress. The increased leptin level promotes adhesion of monocytes, macrophage-to-foam cell transformation, and unfavorable changes in lipid and inflammatory cytokine levels in adipose tissue [28]. All these processes result in the development and destabilization of atherosclerotic plaques [29].

Inflammation plays a crucial role in atherosclerosis, and EAT as a tissue with proinflammatory properties provides a huge contribution to coronary plaque formation [14,17,30,31]. For example, Mazurek et al. showed that plasma inflammatory biomarkers did not adequately reflect local tissue inflammation [17]. Proinflammatory properties of EAT were noted irrespective of clinical variables (diabetes, body mass index, and chronic use of statins or angiotensin receptor enzyme inhibitors/angiotensin II receptor blockers) or plasma concentrations of circulating biomarkers. [17]. Autopsy studies have shown the presence of inflammatory cells in the adventitia in patients with acute coronary syndrome [32]. Furthermore, there was a correlation between the degree of severity of the lesion and the intensification of the inflammatory infiltration in the adventitia [32].

Wang et al. assessed EAT in computed tomography (CT) in patients with and without diabetes, showing that EAT volume (EATV) is higher in diabetic patients and is associated with components of metabolic syndrome, coronary atherosclerosis, and coronary calcium scores [33]. In the Framingham Study, a significant association was found between epicardial fat and coronary artery calcification, which was significant after adjustment for traditional cardiovascular risk factors [34]. Alexopoulos et al. showed that EATV increased significantly with the severity of luminal coronary stenosis and was larger in patients with mixed or noncalcified plaques, compared to patients with calcified plaques or no plaques, indicating the association between EAT and the most dangerous plaque phenotype [35]. This association has been confirmed in other studies [36,37,38]. Yamashita et al. showed that EATV, assessed by CT, is associated with the total coronary plaque burden. Higher EATV was associated with a higher vulnerability of atherosclerotic plaques, based on the evaluation of atherosclerotic plaque composition by intravascular ultrasound imaging (IVUS) [36]. Mazurek et al. made a qualitative assessment of pericoronary adipose tissue (PCAT) using positron emission tomography/computed tomography (PET/CT) among patient with acute coronary syndrome without persistent ST-segment elevation [37] and with stable CAD [38]. In the first study, the inflammatory activity of PCAT, reflected by maximum fluorodeoxyglucose (FDG) uptake, was greater than the activity of adipose tissue in other locations [37]. There was also a correlation between the severity of atherosclerosis and the necrotic core volume of coronary plaque, as assessed by virtual histology IVUS [37]. Similar results were obtained in patients with stable CAD [38]. In another study, EAT thickness was associated with the Thrombolysis in Myocardial Infarction risk score in unstable angina and non-ST-elevation myocardial infarction [39]. Otsuka et al. demonstrated that EATV is associated with the presence of high-risk plaques, the so-called low attenuation plaques in CT, regardless of the presence of abdominal obesity [40]. Patients with higher visceral fat had a greater total plaque volume and a greater level of low-attenuation plaques [40]. Another study assessed the PCAT mean attenuation (PCAT-MA) based on CT as a measure of inflammation in EAT in patients with CAD [41]. PCAT-MA was higher in coronary arteries with plaque compared to vessels without plaque. The lesion-specific PCAT-MA was higher in noncalcified and mixed plaques compared to calcified plaques. These results suggest that lesion-specific PCAT-MA is related to plaque development and vulnerability [41]. In patients with acute myocardial infarction (MI), PCAT attenuation did not differentiate between the coronary segments with and without culprit lesions, but PCAT volume was strongly and independently associated with culprit lesions [42]. In contrast, Goeller et al. showed that PCAT attenuation was increased around culprit lesions compared with nonculprit lesions among patients with acute coronary syndrome [43]. In stable CAD patients, an increase in PCAT attenuation was associated with progression of noncalcified plaque burden and vice versa [44]. Nogic et al. reported that a higher lesion-specific PCAT attenuation baseline may predict in-stent restenosis among patients undergoing elective percutaneous coronary intervention [45]. Altogether, these results confirm that the amount of EAT and/or EAT proinflammatory state correlate with the severity of CAD and plaque vulnerability.

In women with chest pain and angiographically normal coronary arteries, there was a correlation between EAT thickness and reduced coronary flow reserve [46]. Kanaji et al. showed that in CAD patients with a single de novo lesion, PCAT attenuation is significantly associated with global coronary flow reserve [47]. Pasqualetto et al. suggested an association between PCAT attenuation in CT with coronary microvascular dysfunction; however, a significant correlation was found only in patients without severe obstructive CAD [48]. Another study found that EAT might be associated with coronary artery spasms [49]. Further studies are needed to investigate the relationship between EAT/PCAT and coronary microvascular dysfunction and vasospastic angina.

Finally, attention should be paid to the current tendency to study the relationship of CAD not only with the thickness and volume of EAT, but also with its structure and size of adipocytes [50,51]. One study has found that the size and degree of hypertrophy of the epicardial adipocytes are related to CAD severity [51].

## 3. Heart Failure

Among patients with HF, approximately 50% have preserved ejection fraction (HFpEF). HFpEF is a heterogeneous disease with a complex pathogenesis which is not fully understood. This complexity is due to the fact that it can be caused or exacerbated by a variety of comorbidities, including cardiac and extracardiac abnormalities. Thus, the group of patients with HFpEF is very diverse [52,53]. HFpEF is the most common myocardium disorder among obese patients [54]. Savji et al. showed that higher body mass index (BMI) is associated with higher risk of HFpEF than with HF with reduced ejection fraction (HFrEF), and that it was most pronounced among women [55]. Similarly, cardiometabolic features, including insulin resistance, were associated with a higher risk of future HFpEF than with HFrEF [55]. 

Based on the hitherto studies, it can be concluded that: (i) there is an association between EATV and the development of HfpEF; (ii) patients with HFpEF and obesity represent a distinct phenotype of the disease; (iii) EAT thickness or volume may have a greater impact on HFpEF than obesity per se; (iv) EAT participates in the pathogenesis of HfpEF due to EAT’s proinflammatory properties, intensification of fibrosis, and influence on myocardial substrate utilization.

There were several studies that showed a correlation between the severity of left ventricle (LV) diastolic dysfunction and the volume of EAT [56,57,58]. A meta-analysis of 22 studies including 5682 patients also confirmed the correlation between EATV with myocardial diastolic function [59].

Patients with coexisting obesity and HFpEF had a different clinical phenotype than patients with HFpEF without obesity [60]. Obokata et al. compared patients with HFpEF and obesity, HPpEF without obesity, and a non-obese control group without HF [60]. Among obese HFpEF patients, diabetes and sleep apnea were more prevalent, whereas in the non-obese HFpEF patients, atrial fibrillation was more common. Additionally, the obese HFpEF patients had lower concentrations of N-terminal prohormone of brain natriuretic peptide (NT-proBNP), compared to the non-obese cohort. Furthermore, subjects with the obese HFpEF phenotype had increased plasma volume, a higher rate of concentric LV remodeling, greater right ventricular (RV) dilatation, and a higher rate of RV dysfunction. Obese patients also displayed worse exercise capacity, more pronounced hemodynamic abnormalities during exercise, and impaired pulmonary vasodilation. EAT thickness assessed by echocardiography was 20% higher in the obese HF group compared to non-obese HF, and 50% higher compared to the control group [60]. This, along with much greater biventricular hypertrophy, causes an increase in the total heart volume in obese patients, followed by pericardial dilation. If pericardial dilation is insufficient, there is greater coupling between the pericardium, right heart and left heart (interventricular dependance), and pericardial restraint, as showed by greater septal flattening and higher ratio of pressure in the right atrium to pulmonary capillary wedge pressure at rest and during exercise in patients with the obese HFpEF phenotype [60]. Increased EAT thickness may change myocardial substrate utilization, including increased oxygen consumption, impaired oxygen use, and increased dependance on fatty acid oxidation, and thus contributes to a reduction in cardiac reserve and aerobic capacity among obese HFpEF [60,61,62]. In addition, space limitations in the cardiac fossa due to increased EAT thickness may aggravate RV dysfunction and contribute to an increase in intracardiac pressures, especially during exercise [60].

Koepp et al. examined patients with the obese phenotype of HfpEF and divided them into increased EAT thickness (≥9 mm in echocardiography) and normal EAT thickness [63]. It has been demonstrated that obese HfpEF patients with increased EAT thickness have more pronounced hemodynamic derangements at rest and during exercise, including greater elevation in cardiac filling pressures, more severe pulmonary hypertension, and greater pericardial restraint than the obese HFpEF group with normal EAT thickness [63]. Additionally, peak oxygen consumption was 20% lower in patients with increased EAT compared to the normal EAT group [63]. Van Woerden et al. examined EAT in patients with the HF and LV ejection fraction (EF) > 40% (HFpEF and HF with mildly reduced EF) and in the healthy control group using cardiac magnetic resonance (CMR) [64]. It was shown that despite similar BMI, the HF group has significantly higher total and ventricular EATV compared to the control group, and there were no differences in atrial EATV between the groups. These results show that not obesity per se, but rather fat distribution, may contribute to HF development. Additionally, HF patients with AF and/or diabetes had more EAT than HF patients without these disorders. Patients with higher total EATV had higher plasma levels of troponin T, creatine kinase muscle-brain fraction and glycated hemoglobin, and worse kidney function. There were no significant associations between EATV and NT-proBNP concentration. In the HF group, total EATV was positively correlated with LV end-systolic volume and with left and right atrial volume. On the contrary, global longitudinal and circumferential strain were negatively correlated with total EATV [64].

In obese patients with increased plasma volume, the ability of LV to dilate is insufficient, leading to cardiac overfilling and disproportionate increases in cardiac filling pressures. It seems that the inadequate ventricular distensibility is caused by cardiac fibrosis and microvascular rarefaction [65,66]. Moreover, the quantity of fibrosis assessed in CMR is associated with prognosis and outcomes in HFpEF [67]. Obese patients displayed more EAT [8,9] and therefore it seems likely that they were more exposed to cytokines released from the EAT reservoir. As previously noted, in patients with CVDs, the EAT reservoir becomes a site of deranged adipogenesis and a source of proinflammatory factors with deleterious effects on myocardium, including fibrosis. Packer et al. postulated that epicardial fat is a transducer of systemic metabolic disorders and a systemic inflammatory state caused by obesity on the heart [68]. Some researchers pointed to myocardial accumulation of triglycerides and myocardial fibrosis as the main causes of LV diastolic dysfunction [69,70,71]. Additionally, it has been shown that EAT can release vasoactive agents which, via vasa vasorum, reach the microvascular network and may reduce coronary flow reserve [72,73]. One of the most recent studies showed that EAT thickness was of prognostic value in patients with HFpEF, which may be due to increased mechanical restraint and secretion of proinflammatory and proatherogenic adipokines [74]. In contrast, in HFrEF, greater EAT thickness seems to have a protective role, while EAT thinning is associated with a worse prognosis [74]. It is suggested that measuring EAT thickness can be useful to classify patients with or at increased risk of heart failure [75].

Further studies are needed to better understand the influence of EAT on the pathogenesis of HFpEF and EAT’s potential applicability as a target for novel drugs.

## 4. Atrial Fibrillation

AF is the most common arrythmia in the adult population in the world, and its prevalence is increasing. It is estimated that AF affects up to 4% of the population in Australia, Europe, and the USA [76]. The involvement of hemodynamic stress in the pathogenesis of AF is well-documented, and hypertension is the most common risk factor [77]. Valvular diseases also significantly contribute to the development of this arrhythmia [78]. These disorders cause the remodeling of heart chambers, including enlargement of the left atrium (LA) and an increase in LA pressure. Alleviation of hemodynamic stresses can reduce AF’s burden [77,78]. However, there is a large group of patients with AF who do not have hypertension or valvular disease but do have the features of atrial myopathy (LA enlargement, increased LA pressure), as observed in imaging studies [79]. It is known that inflammation is associated with the development of atrial myopathy [80], including both inflammation in course of systemic inflammatory diseases [81,82,83,84] and metabolic disorders accompanied by adipose tissue inflammation [85,86]. The risk of developing AF is especially increased in rheumatoid arthritis [81] and psoriasis [82]. Among the metabolic diseases, special attention should be paid to obesity [85] and diabetes mellitus [86]. In these states, AF’s burden was proportional to the severity of metabolic disorders, such as glycemic control [85,86,87,88].

The literature indicates several potential mechanisms linking EAT with AF, including: (i) proinflammatory status of EAT; (ii) reactive oxygen species (ROS) released from EAT; (iii) fatty infiltration of the atrium; (iv) dysfunction of the autonomic nervous system in EAT.

AF and inflammation are closely associated [80,81,82,83,84,85,86]. EAT can release inflammatory factors and contribute to inflammation and fibrosis in the adjacent myocardium via paracrine signaling. It should be emphasized that EAT has some features of brown adipose tissue, such as the presence of the uncoupling protein-1, which is a thermogenic protein specific to brown adipocytes [89]. These properties are mainly expressed in conditions of health and low oxidative stress [90,91,92]. The healthy EAT is a source of adiponectin, which may reduce inflammation and fibrosis [91,92,93,94]. In obesity, EAT loses its protective properties and becomes a tissue with a proinflammatory profile, subsequently increasing the risk of atrial myopathy and AF [95,96,97,98,99]. Mazurek et al. showed that inflammatory activity of EAT reflected by maximal standardized uptake value of FDG in PET/CT was higher in patients with AF than in the control group and it was not related to BMI [100].

It has been suggested that ROS play an important role in the pathogenesis of AF [101,102]. EAT has been shown to be richer in ROS than other fat depots [103], but at the same time, this effect was reduced by adiponectin [93]. 

It also seems that fatty infiltration into atrial myocardium plays an important role in the pathogenesis of AF, as demonstrated by histological examinations [104]. Fatty infiltration was more pronounced in persistent AF, compared with paroxysmal AF [105]. EATV and fatty infiltration were associated with cardiac conduction abnormalities [106]. It was postulated that EAT can change electrophysiological features and ion currents by cytokine, adipokine, and adipocyte infiltration, causing electrical substrate formation for AF [107]. Opolski et al. showed that increased EATV along the LA assessed by CT was associated with AF after coronary artery bypass grafting [108].

It should be noted that EAT contains significant amounts of ganglionated plexi which are a part of the autonomic nervous system (ANS), which may play a role in the pathogenesis of AF [109,110]. The thickness of the EAT was related to ANS dysfunction [111], and catheter ablation of epicardial fat-reduced cardiac ANS activity, which makes it an interesting therapeutic perspective [112].

There are also several other less-understood potential mechanisms which may explain the involvement of EAT in the pathogenesis of AF, such as the local aromatase effect [113,114,115]. A significant positive correlation was determined between the total aromatase content of EAT and the occurrence/duration of triggered atrial arrhythmias [114]. Further mechanisms are pending investigation.

In clinical terms, the relationship between epicardial adipose tissue and AF is extremely interesting. Obesity is a well-known risk factor for AF [85] and every 1 kg/m^2^ reduction in BMI reduces the risk of AF by about 7% [116]. There is an association between the severity of obesity and the volume and thickness of EAT [8,9]. The relationship between AF and EAT has been investigated in many studies using noninvasive imaging methods such as transthoracic echocardiography, CT, or CMR, showing that: (i) the prevalence of AF is related to the volume/thickness of the EAT; (ii) EAT promotes AF persistence; (iii) higher EATV is associated with lower catheter ablation efficacy.

Results from the Framingham Heart Study involving 2317 patients who underwent CT showed that higher EATV was associated with 40% higher odds of AF and remained significant regardless of traditional risk factors including age, sex, MI, or HF [117]. Interestingly, there was no association between AF prevalence and adipose tissue elsewhere [117].

Chekakie et al. showed a relationship between EATV and both persistent and paroxysmal AF using CT imaging [118]. Patients with persistent AF had a higher EATV than patients with paroxysmal AF or sinus rhythm [118]. The same conclusions were reached by Batal et al., who suggested that EAT can promote AF persistence [119]. Muhib et al. showed similar results in patients with hypertrophic cardiomyopathy using CMR to assess EAT [120].

From a clinical point of view, Wong et al. [121] and Tsao et al. showed a strong relationship between EAT and AF recurrence after catheter ablation. Patients with higher EATV had worse outcomes and early AF recurrence after ablation [121,122].

Finally, two meta-analyses confirmed a relationship between AF and EAT [123,124]. Wong et al. showed the strongest association between persistent AF and EAT, but the association with paroxysmal AF was also significant [123]. Interestingly, the strength of associations between AF with EAT was greater than for between AF and abdominal or overall adiposity [123]. Gaeta et al., based on the analysis of seven imaging studies, demonstrated a 32 mL higher EATV between the AF group and patients with sinus rhythm [124], further indicating that EAT plays a crucial role in AF development and persistence.

## 5. COVID-19

COVID-19 is a complex multisystem infectious disease caused by the SARS-CoV-2 virus, which predominantly affects the lungs [125]. Since COVID-19 was first diagnosed in December 2019, it has caused a significant burden to healthcare systems worldwide. Therefore, there is an urgent need to investigate the pathophysiological mechanisms underlying COVID-19. 

After more than a year of the COVID-19 pandemic, there are reports indicating a potential relationship between EATV and cardiovascular complications of SARS-CoV-2 infection. Systemic inflammation has a central role in the development and progression of COVID-19 [126,127], and there are several studies which have shown that inflammatory and thrombotic biomarkers such as D-dimer or ferritin predict the clinical severity of COVID-19 [128,129,130]. Growing evidence shows that obesity adversely affects the course of mortality due to COVID-19 [131]. In one study, it has been postulated that EAT may have immunomodulatory properties and may be a reservoir for SARS-CoV-2, thus facilitating the spread of the virus and enhancing the inflammatory response [132].

Studies conducted so far on a small number of patients suggest that EATV and attenuation in CT: (i) were independent predictors of severe and unfavorable COVID-19 courses, including death; (ii) may be associated with cardiovascular complications in patients with COVID-19.

Abrishami et al. assessed inflammatory parameters (including CRP) and EAT (volume and density) by CT on admission in 100 patients with COVID-19 [133]. Patients were followed until death or discharge. The mortality rate was 17% and was higher in obese patients. Among laboratory tests, increased lactate dehydrogenase (LDH) and decreased platelet count were significantly associated with death. EATV was similar in patients who died and in those who survived, but EAT density was significantly lower in patients who died (*p* = 0.79 and *p* = 0.008, respectively) [133]. Similar results were obtained by Deng et al. among patients aged 18 to 40. In addition, patients with severe COVID-19 had significantly higher EATV [134].

Iacobellis et al. assessed EAT thickness and density depending on the COVID-19 severity [135]. Patients with most severe course of COVID-19 had significantly greater EAT attenuation than those presenting with mild and moderate COVID-19 (*p* ≤ 0.01), but EAT thickness was similar in all patients [135].

Grodecki et al. examined the relationship of EAT quantified on CT with the severity of pneumonia and adverse outcomes among patients with COVID-19 [136]. The primary outcome was clinical deterioration (intensive care unit admission, invasive mechanical ventilation, or vasopressor therapy) or in-hospital death. Among 109 patients, the primary outcome occurred in 21.1% of patients, and both EATV and attenuation were independent predictors of clinical deterioration or death (*p* = 0.011 and *p* = 0.003, respectively). The severity of pneumonia was also associated with these EAT parameters. Further, there was a correlation between EATV and CRP level and LDH level [136]. The results of the above-mentioned studies indicate that the assessment of EAT by CT may be useful in risk stratification in patients suffering from SARS-CoV-2 infection. However, all these studies were performed in small groups of patients (*n* = 41–109) and studies on larger groups are needed. 

COVID-19 can cause myocardial injury and other cardiovascular complications, including acute myocarditis, pulmonary embolism, or acute heart failure [137,138,139,140,141]. The exact mechanism of heart damage in the course of SARS-CoV-2 infection is not fully understood, but it may occur directly or indirectly, or in both ways [137]. Some cardiovascular complications are asymptomatic during acute infection, but emerging data have reported on post-COVID-19 heart syndrome [142]. It has been suggested that low EAT density in CT may indicate myocardial injury, as it occurs mainly in severe and critical COVID-19 patients [143]. Wei et al. showed in a group of 400 patients with laboratory-confirmed COVID-19 that patients with COVID-19-associated myocardial injury had a history of CVDs, primarily hypertension, diabetes, hypercholesterolemia, and CAD. These patients had higher plasma concentrations of IL-6 and higher risk of adverse in-hospital events (death, invasive mechanical ventilation, admission to an intensive care unit). A chest CT performed on admission showed that these patients also had a higher EATV (139.1 vs. 92.6 cm^2^, *p* = 0.036) and that EATV over 137.1 cm^2^ was a strong independent predictor for myocardial injury in patients with COVID-19 [144]. 

Based on hitherto studies, EATV and EAT density seem to either reflect or affect the overall course of COVID-19, including pulmonary and cardiovascular complication, but more studies are needed to elucidate the mechanisms underlying this association.

## 6. Therapeutic Options to Affect EAT

Since EAT affects the development and progression of CVDs, EAT is a promising therapeutic target in cardiovascular patients. However, none of the therapeutic tools available to date have been specifically developed for EAT. However, it has been shown that (i) lifestyle changes, (ii) bariatric surgery, and (iii) pharmacotherapy can reduce EATV [145,146] by a pleiotropic or an off-target effect (Figure 3).

Both resistance [147,148] and endurance training reduce EATV [147,149,150]. Christensen et al. showed that physical activity can reduce EATV up to 32%, as assessed by CMR [147]. The results of our pilot study in 30 amateur ultramarathon runners are in line with these data [151]. We found that ultrarunners have significantly lower CMR-assessed EATV than the sedentary control group, lower rate of pathologically high levels of plasma IL-6 (>1 pg/mL) and better lipid profile [151]. Therefore, the benefits of regular physical activity to reduce cardiovascular risk may extend beyond the traditional risk factors, as physical activity seems to modulate the EATV and activity.

Another way to reduce EATV is diet [152,153,154,155]. Twenty severely obese subjects followed a 6-month weight-loss program with a very low-calorie diet, achieving a 33% reduction in echocardiographic EAT thickness [155].

There are several small-group studies (*n* = 23 to 65) investigating the effect of bariatric surgery on EAT [156,157,158,159]. Two years after bariatric surgery, EAT thickness was reduced by 31% in a group of 51 operated patients, as assessed by echocardiography [159].

Although the positive effect of lifestyle modifications of EAT and overall cardiovascular health is clear, the compliance remains a concern. Therefore, pharmacotherapy remains a field of great interest regarding the modulation of EATV and function. The following groups of drugs were shown to affect EAT: (i) statins, (ii) antidiabetic drugs, (iii) anti-inflammatory drugs.

Statins have been shown to decrease EATV [160,161]. Alexopoulos et al. demonstrated that statin therapy leads to a reduction in EATV, and intensive therapy was more effective than moderate-intensity therapy in a group of 420 postmenopausal women with hyperlipidemia [160]. There was no correlation between EATV and lipid-lowering effect. Hence, this effect may have been secondary to anti-inflammatory effects of statins [160], which is consistent with the reports on EAT in patients with severe aortic stenosis [162]. Parisi et al. showed a relationship between statin therapy, EAT thickness reduction, and EAT inflammatory status, both in vivo and in vitro [162]. Raggi et al. indicated that statins reduce EAT attenuation in CT independent of their lipid-lowering effect, which indirectly indicates a reduction in EAT inflammation [163]. The clinical benefit of statin therapy in patients with CAD has been known for a long time [164]. However, the range of statin pleiotropic effects is growing, including their anti-inflammatory effects and modulation of EAT [164]. Tawakol et al. showed that statin therapy resulted in rapid, dose-dependent reductions in FDG uptake in PET/CT, representing changes in atherosclerotic plaque inflammation [165]. Even short-term intensive statin therapy significantly reduced the volume of EAT compared to placebo in patients with AF who underwent pulmonary vein isolation [161]. Hence, the anti-inflammatory effect of statins on EAT seems to reduce the risk of atrial myopathy, as demonstrated both in animal and human models [166,167]. The antiarrhythmic effect of statins was also confirmed in randomized trials and was most pronounced in the secondary prevention of AF [168,169,170]. Statins also ameliorate cardiac fibrosis, as demonstrated in animal models of HFpEF [171,172]. The positive effect of statin therapy on LV’s diastolic function was also seen in clinical settings [173,174]. Statin use is also associated with a reduced risk of morbidity and mortality in patients with HF [69,175]. In patients with HFpEF, statin use was associated with reduced mortality [176,177], which was confirmed in meta-analyses [178,179]. On the contrary, no benefit on clinical outcomes was observed in patients with HFrEF [180]. It has been suggested that statins can reduce EAT’s metabolic activity [163]. With the exception of statins, Rivas Galvez et al. showed a significant reduction in thickness of EAT after 6 months of treatment with other lipid-lowering drugs, proprotein convertase subtilisin/kexin type 9 (PCSK-9) inhibitors [181]. All these data may suggest that statins and PCSK-9 inhibitors may exert their pleiotropic effects at least partly through EAT, although the underlying mechanisms of action are yet incompletely understood.

Another group of drugs that affect EAT are antidiabetic drugs, including thiazolidinediones, metformin, sodium-glucose cotransporter 2 (SGLT2) inhibitors, and incretin-based agents. Interest in antidiabetic drugs particularly increased after the publication of the results of the EMPA-REG and LEADER trials, which showed that SGLT2 inhibitors and glucagon-like peptide-1 receptor (GLP-1) agonists could have a cardioprotective effect by a mechanism independent of blood glucose level reduction [182,183]. The older groups of drugs (pioglitazone, metformin) also reduced the risk of cardiovascular complications in patients without diabetes, but with insulin resistance or pre-diabetes [184,185].

Metformin has been the most widely used antidiabetic drug for over 60 years. There is evidence of its anti-inflammatory effects on adipose tissue in diabetic and obese patients [186,187]. The anti-inflammatory effect of metformin has been confirmed by both its anti-aging and antitumor properties [188,189]. Similarly, pioglitazone was shown to reduce mast cells and inflammatory macrophages in adipose tissue [190,191]. These properties seem to be independent of the presence of diabetes mellitus [192]. Ziyrek et al. has shown that metformin monotherapy for 3 months reduced EAT thickness by 10% [193]. The exact mechanism by which metformin interacts with EAT is not clear, but it appears to shift the metabolism into fat oxidation and upregulate the thermogenesis [193,194]. It has recently been shown to be effective against endothelial dysfunction [195]. Chen et al. reported that metformin reduced the secretion of the proinflammatory cytokine, activin A, from epicardial fat [196]. Sardu et al. showed that metformin reduced the inflammatory burden in PCAT and improved prognosis in prediabetic patients with acute MI treated with coronary artery bypass grafting [197]. Metformin through its pleiotropic effect influences the pathogenesis of many cardiovascular diseases [198,199]. One of the mechanisms underlying metformin’s mode of action is the activation of adenosine monophosphate-activated protein kinase, which has an anti-inflammatory effect [198,200]. Studies performed in animal models indicated that SGLT2 inhibitors, GLP-1 agonists, and dipeptidyl peptidase-4 (DPP-4) inhibitors [201,202,203] have similar anti-inflammatory effects in adipose tissue, but further studies are needed to draw firm conclusions.

Thiazolidinediones are another group of drugs which reduce inflammation and the release of proinflammatory cytokines from EAT [204,205,206]. These effects may contribute to the reduced the risk of MI and stroke in patients treated with thiazolidinediones [207,208].

SGLT2 inhibitors were shown to reduce EATV assessed by CMR, among patients with type 2 diabetes, both with and without obesity [209,210,211]. SGLT2 inhibitors also improved the inflammatory status of EAT [209]. They have been documented to be effective against HF and endothelial dysfunction [212,213]. This group of drugs causes a significant reduction in body weight, and one of the mechanisms of action is the stimulation of visceral fat burn [214]. In one study, dapagliflozin caused a reduction in EAT thickness independent of weight loss [215], perhaps thanks to the improvement of EAT cells’ sensitivity to insulin and a reduction in local proinflammatory chemokines secretion [216]. Although SGLT2 inhibitors are relatively new drugs, the first meta-analyses have already confirmed their beneficial impact on EAT [217].

Finally, incretin-based drugs, which include GLP-1 agonists and DPP-4 inhibitors, were also shown to reduce EAT thickness, measured by echocardiography [218] and EATV, assessed by CMR [219]. This effect was mainly weight-loss-dependent [219]. Importantly, despite the reduction in body weight, incretin-based drugs did not reduce the proinflammatory properties of adipose tissue [220,221], although they inhibited the development of atherosclerosis in animal models [222]. It has been shown that GLP-1 receptors are present in EAT, in contrast to subcutaneous adipose tissue [223]. Hence, it has been suggested that GLP-1 agonists affect EAT by stimulating pre-adipocyte differentiation, thermogenesis, and adipocyte browning [224,225]. DPP-4 inhibitors also reduce EAT thickness [226] but can have an adverse effect on its inflammatory status, which may stimulate myocardial fibrosis [227,228,229]. On the other hand, several studies have shown the anti-inflammatory properties of DPP-4 inhibitors [230,231,232]. These studies have suggested that these drugs downregulate the receptor for advanced glycation end-products [233], activate cyclic adenosine monophosphate/protein kinase A signaling and IL-6 production [234], and reduce ROS generation and intercellular adhesion molecule-1 expression [235].

Taking into account the inflammatory nature of atherosclerosis, three anti-inflammatory drugs are important in the context of EAT modulation: (i) canakinumab, (ii) methotrexate, and (iii) colchicine.

Canakinumab is a human anti-interleukin-1-beta (IL-1β) monoclonal antibody which was shown to improve cardiovascular outcomes in over 10,000 patients with a history of CAD and elevated CRP levels in the CANTOS study [236]. Inhibition of the IL-1β/IL-6 signaling cascade with canakinumab led to a significant reduction in cardiovascular risk (nonfatal MI, nonfatal stroke, or cardiovascular death), independent of lipid-level lowering, but with an increased risk of serious infections [236]. Subgroup analysis showed that cardiovascular benefits were achieved due to a reduction in CRP concentration during canakinumab therapy [237].

Methotrexate reduces the proinflammatory effects of IL-6, IL-12, and TNF-alfa and increases the anti-inflammatory effects of IL-10 and IL-1 receptor antagonists [238]. In patients treated with methotrexate for rheumatoid conditions, it reduced the risk of MI by 18%, according to a recent meta-analysis [239]. In the CIRT study that investigated the benefits of methotrexate on outcomes in patients with a history of acute coronary syndrome, MI, and diabetes or metabolic syndrome, no benefits regarding the composite endpoint including nonfatal MI, nonfatal stroke, and cardiovascular death were shown [240]. This is probably due to the fact that elevated CRP levels were not taken into account as an inclusion criterion, in contrast to the CANTOS study.

The efficacy of colchicine in patients with stable CAD was demonstrated by LoDoCo and LoDoCo2 studies [241,242]. In the LoDoCo study, a reduction in a composite endpoint consisting of MI, cardiac arrest, or noncardioembolic stroke was observed in colchicine group, compared to the placebo group [241]. In LoDoCo2 study, a significant reduction in the primary composite endpoint of cardiovascular death, nonprocedural MI, ischemic stroke, or ischemia-driven coronary revascularization was achieved in colchicine group, compared to the placebo [242]. The benefits of adding colchicine to standard therapy were also shown in COLCOT study, which included patients with a history of MI within the last month [243]. The primary efficacy endpoint (a composite of cardiovascular death, resuscitated cardiac arrest, MI, stroke, or urgent hospitalization for angina requiring coronary revascularization) occurred in 5.5% in the colchicine group compared to 7.1% in the placebo group (*p* = 0.02) [242]. Adding colchicine to standard therapy led to a lower risk of ischemic cardiovascular events, compared to the placebo.

Hitherto, there have been no studies evaluating the effects of canakinumab, methotrexate, and colchicine on EAT. Further reports on new drugs affecting EAT and inflammatory mechanisms are expected in the near future. Potential pharmacological therapeutic options are summarized in Table 1.

## 7. Conclusions

EAT is not only an adipose tissue in the histological sense, but is above all a metabolically active tissue, modulating numerous pathophysiological processes in the course of CVDs. In presence of cardiovascular risk factors, the protective properties of EAT are destroyed and it becomes a pro-inflammatory tissue promoting the development and progression of CVDs including CAD, heart failure, arrhythmias, and cardiovascular complications of COVID-19. EAT’s function can be modulated and potentially restored by changing the lifestyle and anti-inflammatory drugs. The development of novel therapies specifically targeting EAT might revolutionize the prognosis in patients with CVD. The search for potential drug targets in EAT is an exciting challenge we currently face.

## Figures and Tables

**Figure 1 biology-11-00355-f001:**
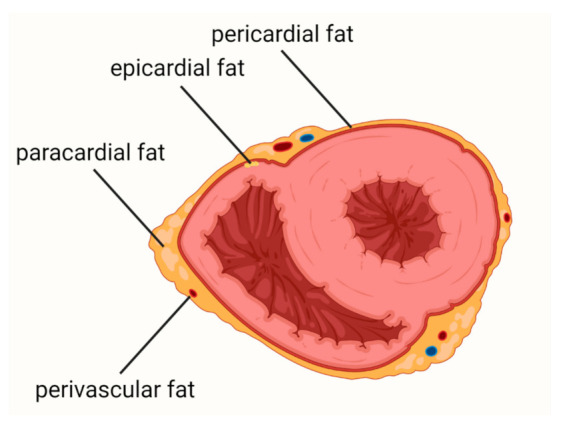
Adipose tissue surrounding the heart.

**Figure 2 biology-11-00355-f002:**
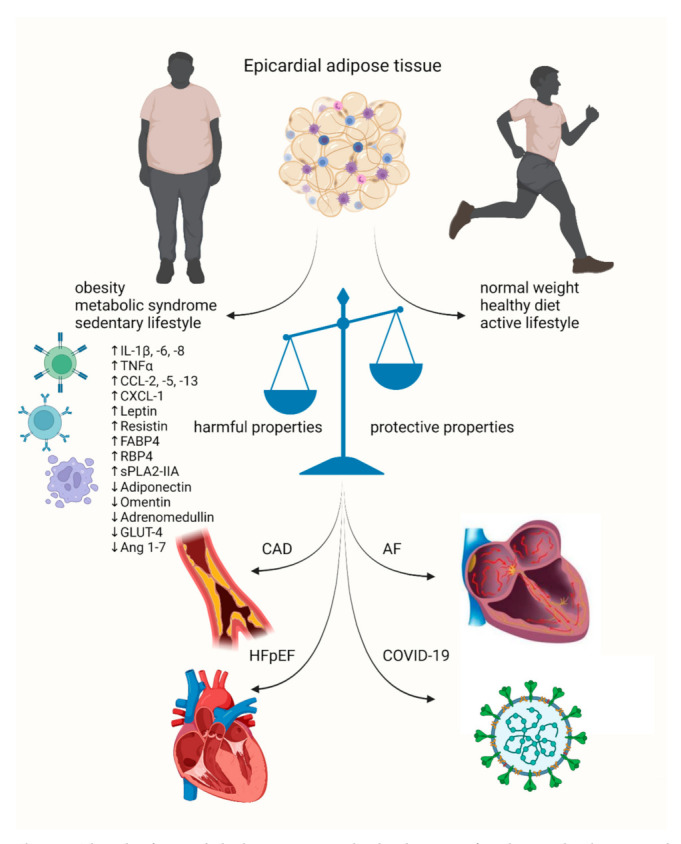
The role of epicardial adipose tissue in the development of cardiovascular diseases and cardiovascular complications in the course of COVID-19. AF—atrial fibrillation; Ang 1–7—angiotensin 1–7; CAD—coronary artery disease; CCL-2, -5, -13—chemokine ligand-2, -5, -13; CXCL-1— chemokine ligand 1; FABP4—fatty acid binding protein 4; GLUT-4—glucose transporter type 4; HFpEF—heart failure with preserved ejection fraction; IL-1β, -6, -8—interleukin-1β, -6, -8; RBP4— retinol binding protein 4; sPLA2-IIA—secretory phospholipase A2; TNFα—tumor necrosis factor α.

**Figure 3 biology-11-00355-f003:**
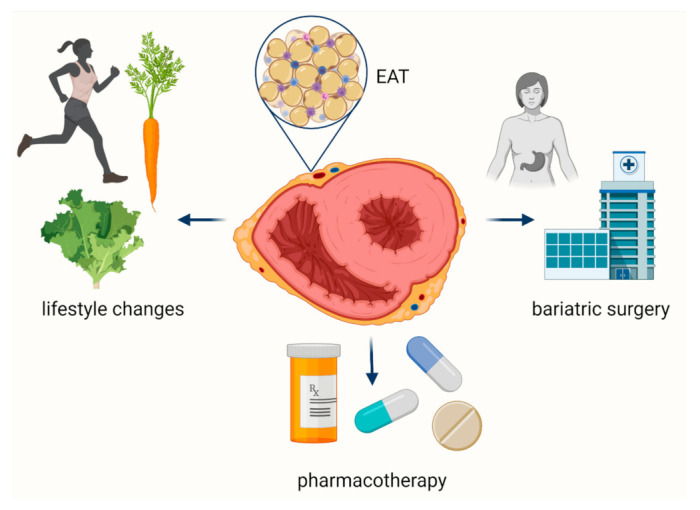
Therapeutic options to affect epicardial adipose tissue.

**Table 1 biology-11-00355-t001:** Pharmacological therapeutic options to affect epicardial adipose tissue.

Pharmacological Therapeutic Options	
Group of Drugs	Potential Mechanisms of Action
Statins	anti-inflammatory [160,162,163,164,165] modulation of EAT [164] ↓ EAT metabolic activity [163]
PCSK-9 inhibitors	unknown
Metformin	anti-inflammatory [186,187,196,197,198,200] ↑ fat oxidation and thermogenesis [193,194] ↓ endothelial dysfunction [195] activation of adenosine monophosphate-activated protein kinase [198,200]
Thiazolidinediones	anti-inflammatory [190,191,204,205,206]
SGLT2 inhibitors	anti-inflammatory [210,217] ↓ endothelial dysfunction [213] stimulation of visceral fat burn [215] ↑ EAT cells sensitivity to insulin [217]
GLP-1 agonists	↑ pre-adipocyte differentiation [225] ↑ thermogenesis [226] ↑ adipocyte browning [226]
DPP-4 inhibitors	anti-inflammatory [231,232,233] downregulation of the receptor for advanced glycation end-products [234] ↑ cyclic adenosine monophosphate/protein kinase A signaling and IL-6 production [235] ↓ ROS generation and intercellular adhesion molecule-1 expression [236]
Canakinumab	anti-inflammatory [237]
Methotrexate	anti-inflammatory [239]
Colchicine	anti-inflammatory [242,243]

EAT—epicardial adipose tissue; DPP-4—dipeptidyl peptidase-4; GLP-1—glucagon-like peptide-1 receptor; ROS—reactive oxygen species; SGLT2—sodium-glucose cotransporter 2.

## Data Availability

Not applicable.

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
