# Peer review of "Role of Epicardial Adipose Tissue in Cardiovascular Diseases: A Review"

_biology, 2022, doi:10.3390/biology11030355_

Round 1
Reviewer 1 Report
This is an interesting review article on the role of epicardial adipose tissue in cardiovascular diseases. The manuscript was written in good English and easy to follow and understand. The flow of the manuscript is also scientifically sound. The only comment that I have is if the authors could provide a summary table to briefly summarize the role of EAT in the development of specific cardiovascular diseases (along with references) and also on the potential therapeutic drugs. This summary table would be very useful to the readers.
Author Response
Dear Reviewer,
We are thankful for the time and effort that you spent to provide in-depth review of our manuscript. We corrected our manuscript according to your suggestions. Our response and corrections are listed below.
The only comment that I have is if the authors could provide a summary table to briefly summarize the role of EAT in the development of specific cardiovascular diseases (along with references) and also on the potential therapeutic drugs. This summary table would be very useful to the readers.
We thank the Reviewer for this comment. The role of EAT in the development of specific cardiovascular diseases is shown in the Figure 2, and as suggested by the Reviewer, we added a table (Table 1.) with drugs and non-pharmacological options that may be active on epicardial fat.
Reviewer 2 Report
The article is devoted to the actual topic of the role of EAT in cardiovascular pathology, including the problem of EAT as a target for therapy. Modern data on the relationship between the volume of EAT and CVD, heart failure and atrial fibrillation are presented in detail. An important chapter is the analysis of therapeutic options to affect EAT.
However, few studies have found that EAT thickness is mediated not only by adipocyte hypertrophy, but also by fibrotic changes in EAT (doi:10.1080/21623945.2019.1701387). At the same time, it was found that adipocyte hypertrophy is associated with the degree of atherosclerosis, but independently of EATV (doi: 10.3390/biomedicines9010064). I believe that this current trend in the study of the relationship not only between the thickness and volume of the EAT, but also its structure, the size of the adipocyte with CVD should be noted.
Minor remarks: In the chapter "Coronary artery disease" lines 90-96 no citations are given, it is better to add them. Correct reference should be made to the statement about the effect of leptin on macrophage-to-foam cell transformation, line 101.
There is some graphical irregularity in Figure 2 - the EAT imbalance and its consequences should be placed under the "fat man" to avoid misinterpreting the figure.
Author Response
Dear Reviewer,
We are thankful for the time and effort that you spent to provide in-depth review of our manuscript. We corrected our manuscript according to your suggestions. Our response and corrections are listed below.
However, few studies have found that EAT thickness is mediated not only by adipocyte hypertrophy, but also by fibrotic changes in EAT (doi:10.1080/21623945.2019.1701387). At the same time, it was found that adipocyte hypertrophy is associated with the degree of atherosclerosis, but independently of EATV (doi: 10.3390/biomedicines9010064). I believe that this current trend in the study of the relationship not only between the thickness and volume of the EAT, but also its structure, the size of the adipocyte with CVD should be noted.
We thank the Reviewer for this comment. We added information on this in the chapter on CAD (lines 166- 169).
Minor remarks: In the chapter "Coronary artery disease" lines 90-96 no citations are given, it is better to add them. Correct reference should be made to the statement about the effect of leptin on macrophage-to-foam cell transformation, line 101.
We added appropriate citations. We added also reference number 28. about the effect of leptin on macrophage to foam cell transformation.
There is some graphical irregularity in Figure 2 - the EAT imbalance and its consequences should be placed under the "fat man" to avoid misinterpreting the figure.
We thank the Reviewer for this comment. We entirely agree with the Reviewer and corrected the figure to make it clearer.
Reviewer 3 Report
I enjoyed reading this narrative review on the association between epicardial adipose tissue and cardiovascular diseases. The manuscript is interesting. The topic is hot.
This reviewer raises some issues and strongly suggests adding and commenting some very up-to-date references closely related to the topic to further enrich this review.
1- I suggest adding in the manuscript a table with drugs that may be active on epicardial fat, describing their respective mechanisms of action.
2- In the chapter dedicated to drugs potentially active on epicardial fat it would be useful to add following references:
a)Metformin has recently been shown to be effective against both endothelial dysfunction and heart failure (Theranostics. 2021 Sep 9;11(19):9376-9396. doi: 10.7150/thno.64706. eCollection 2021. - Heart Fail Rev. 2022 Jan;27(1):337-344. doi: 10.1007/s10741-020-09987-z.)
b) The anti-inflammatory action of metformin is confirmed by both its anti-aging and anti-tumor action (Diabetes Res Clin Pract. 2020 Feb; 160:108025. doi: 10.1016/j.diabres.2020.108025. - ESMO Open. 2017; 2(2): e000132. Published online 2017 May 2. doi: 10.1136/esmoopen-2016-000132.).
c) Likewise, gliflozines have been documented to be effective against HF and endothelial dysfunction. (Heart Fail Rev. 2021 Aug 11. doi: 10.1007/s10741-021-10157-y. - 2021 May;52(5):1545-1556. doi: 10.1161/STROKEAHA.120.031623.).
3- Very recently a review on epicardial adipose tissue in subjects with or at risk of HF (Heart Fail Rev. 2021 Oct 20. doi: 10.1007/s10741-021-10160-3.). The authors should add the above reference in the text.
4- The paper should be reviewed by a native English speaker.
Author Response
Dear Reviewer,
We are thankful for the time and effort that you spent to provide in-depth review of our manuscript. We corrected our manuscript according to your suggestions. Our response and corrections are listed below.
I suggest adding in the manuscript a table with drugs that may be active on epicardial fat, describing their respective mechanisms of action.
We thank the Reviewer for this comment. We added a table (Table 1.) with drugs and non-pharmacological options that may be active on epicardial fat. This is just a general table, without describing the mechanisms of action, as the exact mechanisms of action on the EAT are still poorly understood.
In the chapter dedicated to drugs potentially active on epicardial fat it would be useful to add following references:
a)Metformin has recently been shown to be effective against both endothelial dysfunction and heart failure (Theranostics. 2021 Sep 9;11(19):9376-9396. doi: 10.7150/thno.64706. eCollection 2021. - Heart Fail Rev. 2022 Jan;27(1):337-344. doi: 10.1007/s10741-020-09987-z.)
- b) The anti-inflammatory action of metformin is confirmed by both its anti-aging and anti-tumor action (Diabetes Res Clin Pract. 2020 Feb; 160:108025. doi: 10.1016/j.diabres.2020.108025. - ESMO Open. 2017; 2(2): e000132. Published online 2017 May 2. doi: 10.1136/esmoopen-2016-000132.).
- c) Likewise, gliflozines have been documented to be effective against HF and endothelial dysfunction. (Heart Fail Rev. 2021 Aug 11. doi: 10.1007/s10741-021-10157-y. - 2021 May;52(5):1545-1556. doi: 10.1161/STROKEAHA.120.031623.).
We thank the Reviewer for this comment. We added these publication to the reference.
3- Very recently a review on epicardial adipose tissue in subjects with or at risk of HF (Heart Fail Rev. 2021 Oct 20. doi: 10.1007/s10741-021-10160-3.). The authors should add the above reference in the text.
We thank the Reviewer for this comment. We added this publication to the reference.
The paper should be reviewed by a native English speaker.
We thank the Reviewer for this comment. We tried to correct the manuscript in terms of language.
Round 2
Reviewer 3 Report
No further comments.
Author Response
Dear Reviewer,
We are grateful for your time and all comments.